# Verification of Safety and Efficacy of Sleeve Gastrectomy Based on National Registry by Japanese Society for Treatment of Obesity

**DOI:** 10.3390/jcm12134303

**Published:** 2023-06-27

**Authors:** Shinichi Okazumi, Takashi Oshiro, Akira Sasaki, Hisahiro Matsubara, Ichiro Tatsuno

**Affiliations:** 1Japanese Society for Treatment of Obesity, Tokyo 113-0033, Japan; sokazumiyrsh@sakura.med.toho-u.ac.jp (S.O.);; 2Department of Surgery, Toho University Sakura Medical Center, 564-1, Shimoshizu, Sakura 285-8741, Japan

**Keywords:** national registry, bariatric surgery, sleeve gastrectomy, safety, efficacy

## Abstract

In Japan, bariatric surgical treatment was started in 1982. The Japanese Society for Treatment of Obesity (JSTO) was established in 2007, and then, JSTO started the national registry of bariatric surgery cases and multidisciplinary educational program. A total of 44 facilities registered 4055 bariatric surgical cases until 2021. In this study, the purpose is to clarify the indication, the safety and the effectiveness of the sleeve gastrectomy using national registry database compiled by JSTO. Preoperative BMI ranged from 27.6 to 90.7 kg/m^2^, and the mean value was 42.7. With regard to gender, men/women was 1/1.3. Age was 42.2 as mean. As preoperative comorbidities, DM ratio was 54.4% of the patients, hypertension 64.5%, dyslipidemia 65.1%, and sleep apnea syndrome 69.8%. As an operation method, laparoscopic method was conducted in 99.7% of the cases. The intraoperative incidence rate was 0.9%. Conversion rate to open method was 1.1%. Postoperative morbidity ratio was 5.6%, and mortality was 0%. Reoperations were performed in 1.5% of the cases. Postoperative hospital stay was 5 days in median value. Body weight loss was 27.6 kg in the mean value after follow-up days of 279 ± 245. As the effect on the preoperative metabolic comorbidities, DM has improved in 82.9% of the cases, hypertension 67.9% and dyslipidemia 66.6%. In conclusion, using JSTO database, we evaluated the indication, postoperative complications and weight loss effect of sleeve gastrectomy in Japan. Regarding the evaluation of the effect on preoperative comorbidities, future follow-up based on more detailed criteria was considered to be necessary.

## 1. Introduction 

Surgical treatment for obesity began in the 1950s. Sleeve gastrectomy, which was developed in the 2000s, has become popular worldwide with the progress of laparoscopic surgery and is currently the most performed bariatric surgical procedure [1]. In this surgery, the weight loss effect and the metabolic improvement effect are remarkably observed. Therefore, this surgery has been re-recognized as a metabolic surgery, and the number of surgical cases is increasing [2,3]. The low invasiveness, weight loss effect and metabolic improvement effect of this procedure have been highly evaluated, and research on the factors that influence the effect has progressed [4,5,6]. On the other hand, with the spread of this technique, it is important to perform it safely for short-term and long-term results. A management system is in place for this purpose. In the U.S., ACS (American College of Surgeons) and ASMBS (American Society for Metabolic and Bariatric Surgery) have their own education and certification systems to ensure safety [7,8]. In Japan, bariatric surgical treatment was started in 1982. The first case was performed using open method. After that, the Japanese Society for Treatment of Obesity (JSTO) was established in 2007 due to an increase in diabetes and an increase in the trend toward obesity among adult men, and then, JSTO started national registry of bariatric surgery cases and multidisciplinary educational program in order to facilitate surgical treatment of obesity in Japan (Table 1). 

The national registry database by JSTO aims to ensure safety and efficacy in order to facilitate increased utilization of bariatric surgery. The database committee decided input items at the time of its start, and the analyzed results were shared with the board of directors and the society members through annual reports published at each academic society. Before the establishment of JSTO, total cases were 186, and half of them were performed using open method. After the establishment of JSTO, the numbers were increasing and reached 4055 in total in 2021 (Figure 1). Most operation methods were performed laparoscopically. Additionally, recently sleeve gastrectomy was indicated for many cases according to the prevalence of laparoscopic method. In Japan, for obesity surgery, only sleeve gastrectomy was allowed for insurance medical treatment. If the main purpose was to treat diabetes, bypass surgery was often recommended to the patient, but in this case, it would be self-financed. In Japan, before 2008, half were gastric bypass and the other half were banding method and sleeve gastrectomy. After 2008, sleeve gastrectomy was mostly performed in 3240 cases (83.7%), and the next was sleeve bypass method [9] (Figure 2). Under such background, in this study, we verified the safety and efficacy of sleeve gastrectomy in Japan based on national registration by JSTO.

## 2. Subjects and Methods 

### 2.1. JSTO Database and Subjects 

The study was based on the database registered by JSTO. Each member facility of JSTO was issued an ID and password. The database includes gender, age, weight, BMI, comorbidities (including diabetes, hypertension, dyslipidemia, sleep apnea and other complications), treatment efficacy rate, date, surgical technique, laparoscopy, change to open surgery, reoperation, surgical complications, postoperative complications, postoperative hospital stay period, postoperative follow-up period, weight loss (kg), diabetes improvement, hypertension improvement and dyslipidemia improvement for each registered patient. To avoid complication for registration at each facility, it emphasizes on the ease of entering input terms. The registrations were uploaded after surgery at arbitrary timing. Therefore, follow-up period was up to 1 year at most registered institutions. Determination of treatment response was based on weight loss and improvement in comorbidities at enrollment, and specific criteria were entrusted to each site to simplify the process. Physician standards and facility standards were considered as ethical guidelines in accordance with health insurance medical fees or clinical research. The data were anonymized by each facility. Authors were members of the database committee of JSTO and directly parsed the database. In this study, the aim is to clarify the indication, the safety and the effectiveness of the sleeve gastrectomy using national registry database compiled by JSTO. The number of the subjects were 3240. The period of the surgery was from 2008 to 2021. 

### 2.2. Methods 

Indications of surgery, the safety of the surgery and the outcome of bariatric and metabolic effects were evaluated. The evaluated factors were preoperative BMI, age, gender proportion and coexisting comorbidities as indications of surgery. BMI and age were estimated by mean ± SD. Preoperative comorbidities input was either of the presence or the absence, so that they were evaluated by the positive rate (%). As the safety of the surgery, laparoscopic procedure, change to open surgery, reoperation, surgical complications and postoperative complications were estimated by positive rate (%). Additionally, postoperative hospital stay period was evaluated by median value. As the outcome of bariatric effects, body weight loss (kg) and the postoperative follow-up period were evaluated by mean ± SD. As the outcome of metabolic effect, diabetes, hypertension and dyslipidemia were evaluated. Postoperative improvements were input as the presence or the absence, and they were evaluated by the positive rate (%). 

## 3. Results

### 3.1. Indications of Surgery

Indications of surgery were determined according to the JSTO statement, IFSO-APC standards or Japanese insurance medical treatment requirements. The evaluated factors were preoperative BMI, age, gender and the coexisting morbidities. Preoperative BMI was 27.6 to 90.7 kg/m^2^ (mean ± SD 42.7 ± 8.0) (Figure 3).

As gender, men were 1385 (42.7%), and women were 1855 (57.3%). Age was 42.2 ± 10.4 (mean ± SD) (M 42.5 ± 9.5, F 42.5 ± 11.1). As preoperative comorbidities, the combined ratio of DM was 54.4%, hypertension 64.5%, dyslipidemia 65.1% and sleep apnea syndrome (SAS) 69.8%. 

### 3.2. Evaluation of the Safety of the Surgery

As an operation method, laparoscopic methods were performed in 99.7% of the cases. The intraoperative incidence rate was 0.9% which included difficulties (*n* = 4; 0.12%), organ injury (*n* = 11; 0.34%), bleeding (*n* = 6; 0.19%), staple trouble (*n* = 2; 0.06%), hypercapnia (*n* = 2; 0.06%) and others (*n* = 4; 0.12%). Conversion rate to open from laparoscopic methods was 1.1%. As postoperative complications, bleeding (*n* = 57; 1.76%), SSI (*n* = 19; 0.59%), pneumonia (*n* = 18; 0.56%), stenosis (*n* = 17; 0.52%), anastomotic leakage (*n* = 10; 0.31%), mental disorder (*n* = 7; 0.21%), GERD (*n* = 6; 0.19%), infection (*n* = 6; 0.19%), heart disease (*n* = 5; 0.15%), thrombosis (*n* = 5; 0.15%), liver dysfunction (*n* = 4; 0.12%), rhabdomyolysis (*n* = 3; 0.09%), renal failure (*n* = 3; 0.09%), dehydration (*n* = 3; 0.09%) and others (*n* = 20; 0.62%) were registered. Reoperations were performed in 1.5% of the cases due to bleeding (*n* = 22; 0.68%), anastomotic leakage (*n* = 5; 0.15%), stenosis (*n* = 3; 0.09%), abscess (*n* = 1; 0.03%) and others (*n* = 17; 0.52%). Postoperative complication rate was 5.6% in total, and mortality was 0%. Postoperative hospital stay was 5 days as the median value. 

### 3.3. Evaluation of the Outcome of Bariatric and Metabolic Effects 

Body weight loss was 27.6 ± 15.5 (mean ± SD) kg in the follow-up days of 279 ± 245 after surgery.

As the effect on the preoperative metabolic comorbidities, DM improved in 82.9% of the cases, hypertension in 67.9% and dyslipidemia in 66.6% after 279 ± 245 days after surgery.

## 4. Discussion

The obese population (BMI > 30) in Japan is about 3%, which is one of the lowest in the world [10]. On the other hand, the diabetic population is on the rise in Japan [11], and the relationship between obesity and diabetes has also been reported [12]. In particular, Asians are said to be underweight and prone to diabetes compared to Westerners [13]. In the 2000s, the metabolic improvement effect of bariatric surgery was recognized, and it came to be called ‘metabolic surgery’ [14]. In particular, it is highly effective in improving diabetes, and its indications have been determined by academic societies worldwide [15,16,17]. A lower BMI indication has been shown for Asians [18].

In Japan, JSTO was established in 2008 with the aim of safe increase in surgery while feeling the need to expand bariatric surgical treatment. Under the leadership of the society, we have promoted the guideline that created the surgical case registration, the multi-professional education program and the facility certification business. As of 2008, there were less than 200 surgical cases, which rapidly increased thereafter, reaching over 4000 cases in 2021 after being covered by health insurance in 2014. In JSTO, we have reported safety and efficacy of surgical cases at each annual meeting, and both have shown good results and have been improving year by year. In the present analysis, the intraoperative incident rate was 0.9%, and the postoperative complication rate was 5.6%. The reoperation rate was 1.5%, and there were no deaths, and it was considered that the surgical treatments were performed safely. The operative safety were feasible compared to previous reports, which showed 6.5–14% postoperative morbidities [19,20]. The median postoperative hospital stay was 5 days. In Japan, in order to confirm safety, there were many facilities where patients were discharged from the hospital in 3 to 5 days even if the surgery went smoothly. In the future, it is expected to be shorten.

Looking at the improvement in the follow-up days of 279 ± 245 after surgery, weight loss effect 27.6 ± 15.5 kg and the improvement of comorbidities was observed in the majority of patients, with diabetes mellitus 82.9%, hypertension 67.9% and dyslipidemia 66.6%. The effect was prominent in diabetes. The efficacy in long-term outcomes has also been reported by multiple institutions [21,22,23]. As described above, the indication and outcome of sleeve gastrectomy has been shown and verified that it is safely administered in Japan, which is located in East Asia and is prone to suffering from diabetes. The laparoscopy rate was 99.7%, and the conversion to open rate was 1.1%. Minimally invasive procedures are thought to contribute to safety. On the other hand, the registry by JSTO were uploaded after surgery at arbitrary timing. Therefore, follow-up period was up to 1 year at most registered institutions. Therefore, there was some range in the registered bariatric and metabolic effects of cases according to the timing of registration. In addition, preoperative comorbidity input was either of the presence or the absence, and postoperative improvement was also input of the presence or the absence. Improvement indicators were based on the criteria of each facility; therefore, there were limitations in the detailed analysis of this study. In the registry, the characteristic postoperative complications of sleeve gastrectomy, such as a stump of suture failure, twisting, stenosis and reflux esophagitis, have also been registered. It is considered necessary to devise ways to avoid these problems and to further consider treatments to be taken at the time of onset. In addition, multi-disciplinary education [24], facility certification, surgical case registration, and verification of safety effectiveness are considered important, and we would like to continue them in the future. Additionally, in this study, there were some limitations due to which it was considered impossible to construct a specific predictive index. As a future task, we considered the direction of building a follow-up registry system.

## 5. Conclusions

In conclusion, using JSTO database, we evaluated the indication, postoperative complications and weight loss effect of sleeve gastrectomy in Japan. Regarding the evaluation or prediction of the effect on preoperative comorbidities, future follow-up analysis based on more detailed criteria is considered to be necessary.

## Figures and Tables

**Figure 1 jcm-12-04303-f001:**
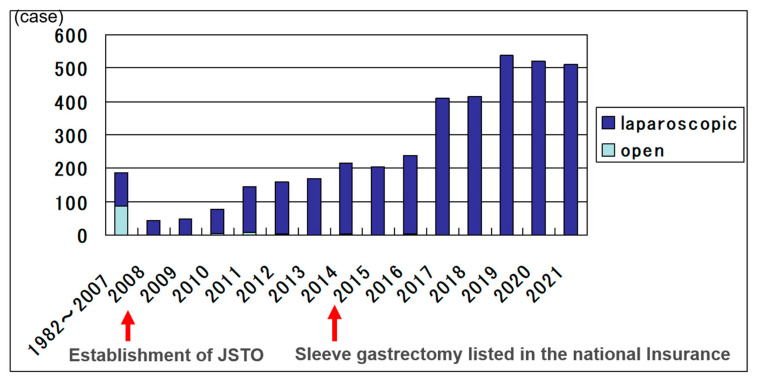
Bariatric surgery cases in Japan. National registry of the bariatric surgery by 44 facilities including 4055 cases: 1982–2007 (*n* = 186) and 2008–Dec. 2021 (*n* = 3869).

**Figure 2 jcm-12-04303-f002:**
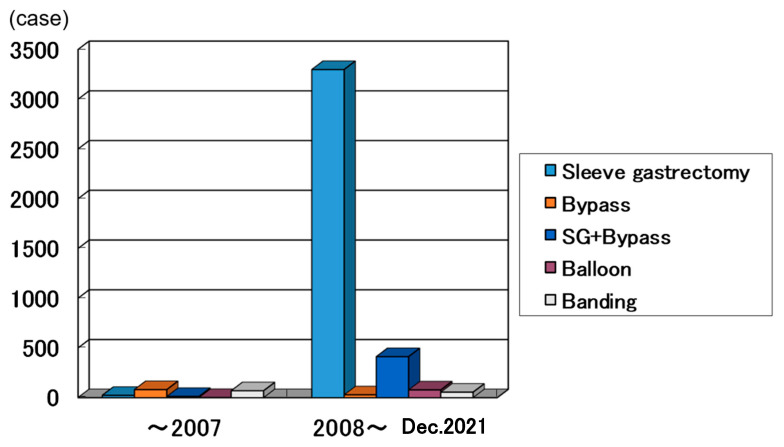
Operation methods (44 facilities; 4055 cases) in Japan. After 2008, sleeve gastrectomy was mostly performed in 3240 cases (83.7%).

**Figure 3 jcm-12-04303-f003:**
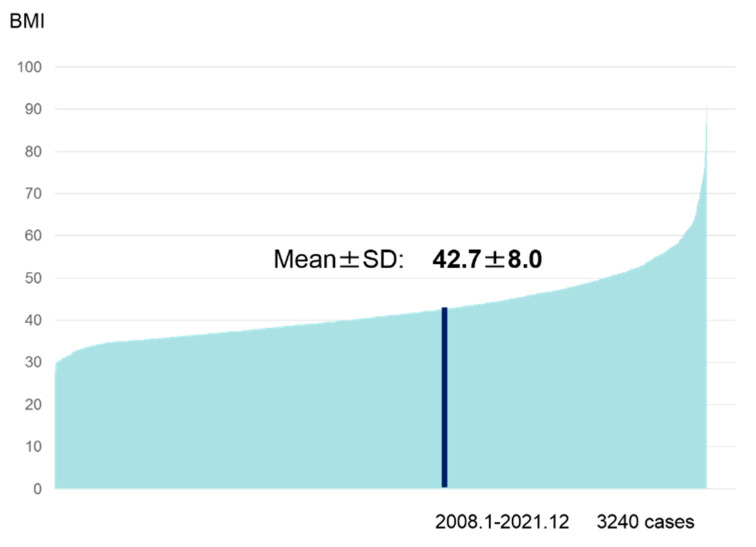
Preoperative patient’s body mass indices. The value data ranged from 27.6 to 90.7 kg/m^2^ (mean ± SD:42.7 ± 8.0).

**Table 1 jcm-12-04303-t001:** The activities of Japanese Society for Treatment of Obesity (JSTO).

2007	Establishment of Japanese Society for Treatment of Obesity (JSTO)
2008	1st general meeting of JSTO
2010	Statement for safe and excellent bariatric surgery
2012	Start of national registry of bariatric surgery
2013	Guideline for safe and excellent bariatric surgery in Japan
2014	Sleeve gastrectomy in national health insurance system
2016	Publishing of handbook of metabolic surgery
2021	Total registration of bariatric surgery over 4000 cases
2022	Consensus statement on the bariatric and metabolic surgery for Japanese patients with severe obesity with T2DM

## Data Availability

Our data used in this study is available on request from the corresponding author.

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
