# Peer review of "Verification of Safety and Efficacy of Sleeve Gastrectomy Based on National Registry by Japanese Society for Treatment of Obesity"

_jcm, 2023, doi:10.3390/jcm12134303_

Round 1
Reviewer 1 Report
This a nice and simple manuscript on the safety and efficacy of sleeve gastrectomy in Japanese population, aimed to to clarify the indication, the safety and the effectiveness of the sleeve gastrectomy.
Comments:
3.2. Indications of surgery
1.- Authors indicate that preoperative BMI JSTO. Preoperative BMI was ranged from 27.6 to 90.7 kg/m2, and the mean value was 42.7.
I guess overweight, and class 1, 2 and 3 subjects were included in the studied population. In my opinión, authors should abound in the corresponding figures.
2.- 3.4. Evaluation of the outcome of bariatric and metabolic effects
Body weight loss was 27.6 ± 15.5 (average ± SD)kg in the follow up days of 279 ± 245 after surgery. As the effect on the preoperative metabolic comorbidities, DM improved in 82.9% of the cases, hypertension 67.9% and dyslipidemia 66.6% after 279 ± 245 days after surgery.
It is clear that the rates of improvement of DM, hypertension and dyslipidemia are very relevant. However, it is not clear what the improvement consists of. Are they simple improvement of baseline values tending towards normality or reversion to normal figures?
On the other hand, authors describe that a body weight loss was 27.6 ± 15.5 (average ± SD)kg was achieved after the intervention, which, in my opinión, implies the existence of a great efficacy inter-individual variability, as usually observed with any anti-obesity intervention. Thus, an exploratory análisis data could provide information about about specific subpopulations in which the intervention is more or less useful
Author Response
Response to reviewer 1
We deeply thank you for your consideration and appreciate your instructive comments. We revised our article according your comments. We added descriptions and corrections.
Comments and Suggestions for Authors
This a nice and simple manuscript on the safety and efficacy of sleeve gastrectomy in Japanese population, aimed to to clarify the indication, the safety and the effectiveness of the sleeve gastrectomy.
Comments:
3.2. Indications of surgery
1.- Authors indicate that preoperative BMI JSTO. Preoperative BMI was ranged from 27.6 to 90.7 kg/m2, and the mean value was 42.7.
I guess overweight, and class 1, 2 and 3 subjects were included in the studied population. In my opinión, authors should abound in the corresponding figures.
⇒ Preoperative subject’s BMI was 27.6 to 90.7 kg/m2(mean±SD 42.7±8.0).
A figure showing the weight distribution of the target cases was added (figure 3).
2.- 3.4. Evaluation of the outcome of bariatric and metabolic effects
Body weight loss was 27.6 ± 15.5 (average ± SD)kg in the follow up days of 279 ± 245 after surgery. As the effect on the preoperative metabolic comorbidities, DM improved in 82.9% of the cases, hypertension 67.9% and dyslipidemia 66.6% after 279 ± 245 days after surgery.
It is clear that the rates of improvement of DM, hypertension and dyslipidemia are very relevant. However, it is not clear what the improvement consists of. Are they simple improvement of baseline values tending towards normality or reversion to normal figures?
⇒ For the benefit to avoid complication at registration of each facility, it emphasizes the ease of entering input terms. The registrations were uploaded after surgery at any time. Therefore, follow-up period was up to 1 year at most registered institutions. Determination of treatment response was based on weight loss and improvement in comorbidities at enrollment, and specific criteria were entrusted to each site to simplify the process. Therefore, the distinction between ‘simple improvement of baseline values tending towards normality or reversion to normal figures’ was limited from this database. We added the description(page.4)
On the other hand, authors describe that a body weight loss was 27.6 ± 15.5 (average ± SD)kg was achieved after the intervention, which, in my opinión, implies the existence of a great efficacy inter-individual variability, as usually observed with any anti-obesity intervention. Thus, an exploratory análisis data could provide information about about specific subpopulations in which the intervention is more or less useful
⇒ Body weight loss was 27.6±15.5 (mean±SD)kg in the follow up days of 279±245 after surgery. The registrations were uploaded after surgery at arbitrary timing. Therefore, follow-up period was up to 1 year at most registered institutions. Therefore, there was some range in the registered bariatric and metabolic effects of cases according to the timing of registration. We added the description in the discussion(page.7).

Reviewer 2 Report
Prevention and treatment of obesity are part of the health pillars in developed countries. The complications of this pathology pose a great risk to the life of the population, and a great economic expense for society. The authors focus on the evaluation of one of the most widely used surgical treatments worldwide, sleeve gastrectomy, from a safety and efficacy perspective.
Some important points need to be clarified for a better understanding of the study:
- In methodology section, it is specified that those responsible for the database under study is maintained by the Japanese Society for the Treatment of Obesity (JSTO), and that its results are shared with the board of directors, board of trustees, and surgery subcommittee through an annual report. Are the authors part of these groups? Were these data extracted from the annual report or were they obtained directly from the database? It is necessary to specify in this section how the data was obtained in a more specific way, and if the database (not the annual report) is accessible to the public.
- Based on the methodology and the results presented, it is reported that the database records many variables of each of the reported cases, but in this article only some of these were studied, such as preoperative BMI, coexisting comorbidities, intraoperative incidents, post-operative complications, reoperation, morbidity, mortality, hospital stay and the effect on the body weight and the metabolic comorbidities. What was the reason why recorded variables such as sex, age, percentage of abdominal fat were not considered in the present study? All these variables provide greater richness when characterizing the sample, allowing data segmentation, and checking whether any of these variables is important when evaluating the efficacy and safety of this type of surgical strategies. On the other hand, have analytical parameters of these patients been recorded, such as the lipid profile or indicators of glycemic control? These variables would grant greater objectivity when determining the improvement of the patients.
- Continuing with the previous observation, the results can be enriched by including the sample characterization variables. You can work in greater depth with the database to establish possible relationships or dependencies between the study variables. A descriptive statistical analysis, together with an inferential and predictive analysis would greatly enrich the work presented. It is recommended to include this analysis in all sections of the results.
- An important point is to determine what was the indication for surgical intervention. Section 3.2 of the results specifies the BMI range of the individuals included in the study (27.6 and 90.7 kg/m2). Analyzing this range, patients who were overweight have undergone surgery. Have the indications for bariatric surgery been recorded in the database? If so, what was the indication in those cases in which the patients were overweight, the intervention was aimed at the treatment of other metabolic or oncological pathologies?
- In the discussion section, the authors mention some results obtained, but do not discuss them with other interventions, with the results obtained in other countries, with the recommendations made by the JSTO. Without this in-depth analysis, the study's conclusions, determining that the safety and efficacy of sleeves gastrectomy are verified using national registry database compiled by JSTO, lack compelling evidence. What does the JSTO and the authors identify as safe and efficient procedures? What bibliography are they based on to determine them?
On the other hand, it is necessary to address some minor observations to improve the presentation of the article:
- In section 2 subjects and methods it is necessary to include a small paragraph informing how the data will be treated, that central tendency measures will be used with their corresponding deviation or that in some cases frequencies will be used.
- In figure 1 (page 2) and figure 2 (page 3) it is necessary to establish the characteristics of the ordinate axis, indicating that it is the number of registered cases.
- On page 3, the final sentence of section 3.1, just below figure 2, expresses the same information as that presented in the previous paragraph. It is necessary to remove it.
- On page 4, in section 3.2, it is specified that the mean BMI was 42.7. Deviation measures need to be included.
Best regards
Author Response
Response to reviewer 2
We deeply thank you for your consideration and appreciate your instructive comments. We revised our article according your comments. We added descriptions and corrections.
Comments and Suggestions for Authors
Prevention and treatment of obesity are part of the health pillars in developed countries. The complications of this pathology pose a great risk to the life of the population, and a great economic expense for society. The authors focus on the evaluation of one of the most widely used surgical treatments worldwide, sleeve gastrectomy, from a safety and efficacy perspective.
Some important points need to be clarified for a better understanding of the study:
- In methodology section, it is specified that those responsible for the database under study is maintained by the Japanese Society for the Treatment of Obesity (JSTO), and that its results are shared with the board of directors, board of trustees, and surgery subcommittee through an annual report. Are the authors part of these groups? Were these data extracted from the annual report or were they obtained directly from the database? It is necessary to specify in this section how the data was obtained in a more specific way, and if the database (not the annual report) is accessible to the public.
⇒ Authors were members of the database committee and only the committee directly parsed the database. The data were anonymized by each facility and not accessible to the public. We added the description(page.4).
- Based on the methodology and the results presented, it is reported that the database records many variables of each of the reported cases, but in this article only some of these were studied, such as preoperative BMI, coexisting comorbidities, intraoperative incidents, post-operative complications, reoperation, morbidity, mortality, hospital stay and the effect on the body weight and the metabolic comorbidities. What was the reason why recorded variables such as sex, age, percentage of abdominal fat were not considered in the present study? All these variables provide greater richness when characterizing the sample, allowing data segmentation, and checking whether any of these variables is important when evaluating the efficacy and safety of this type of surgical strategies. On the other hand, have analytical parameters of these patients been recorded, such as the lipid profile or indicators of glycemic control? These variables would grant greater objectivity when determining the improvement of the patients.
⇒The age and gender of the subjects were described in detail (page 5). As gender, men were 1385(42.7%) and women were 1855(57.3%). Age was 42.2±10.4(mean± SD)(M 42.5±9.5,F 42.5±11.1). ‘Percentage of abdominal fat ‘was not analyzed as it was not a required registry item.
For the purpose of simplifying the input, preoperative comorbidity input was either of the presence or absence, and postoperative improvement was also input of presence or absence. Improvement indicators were based on the criteria of each facility, therefore, there was limitation in detailed analysis. We added the description in the discussion (page.7).
- Continuing with the previous observation, the results can be enriched by including the sample characterization variables. You can work in greater depth with the database to establish possible relationships or dependencies between the study variables. A descriptive statistical analysis, together with an inferential and predictive analysis would greatly enrich the work presented. It is recommended to include this analysis in all sections of the results.
⇒ The registry by JSTO were uploaded after surgery at arbitrary timing. Therefore, follow-up period was up to 1 year at most registered institutions. Therefore, there was some range in the registered bariatric and metabolic effects of cases according to the timing of registration, and the improvement indicators of metabolic effect were based on the criteria of each facility, therefore, there were limitation in detailed analysis, and then in this study, it was considered impossible to construct a specific predictive index. As a future task, we considered the direction of building a follow-up registry system. We added the description in the discussion (page.7).
- An important point is to determine what was the indication for surgical intervention. Section 3.2 of the results specifies the BMI range of the individuals included in the study (27.6 and 90.7 kg/m2). Analyzing this range, patients who were overweight have undergone surgery. Have the indications for bariatric surgery been recorded in the database? If so, what was the indication in those cases in which the patients were overweight, the intervention was aimed at the treatment of other metabolic or oncological pathologies?
⇒ Indications for surgery are determined according to the JSTO statement, IFSO-APC standards, and Japanese insurance medical treatment requirements. In Japan, only sleeve gastrectomy was used for insurance medical treatment. If the main purpose was to treat diabetes, bypass surgery were recommended to the patient, but in this case, it would be self-financed. We added these details(page.3,5).
- In the discussion section, the authors mention some results obtained, but do not discuss them with other interventions, with the results obtained in other countries, with the recommendations made by the JSTO. Without this in-depth analysis, the study's conclusions, determining that the safety and efficacy of sleeves gastrectomy are verified using national registry database compiled by JSTO, lack compelling evidence. What does the JSTO and the authors identify as safe and efficient procedures? What bibliography are they based on to determine them?
⇒ The postoperative safety were feasible compared to previous reports. We added references [19][20], and described in the discussion(page.7).
On the other hand, it is necessary to address some minor observations to improve the presentation of the article:
- In section 2 subjects and methods it is necessary to include a small paragraph informing how the data will be treated, that central tendency measures will be used with their corresponding deviation or that in some cases frequencies will be used.
⇒ We added the description(page 5).
- In figure 1 (page 2) and figure 2 (page 3) it is necessary to establish the characteristics of the ordinate axis, indicating that it is the number of registered cases.
⇒ We corrected them.
- On page 3, the final sentence of section 3.1, just below figure 2, expresses the same information as that presented in the previous paragraph. It is necessary to remove it.
⇒ We corrected them.
- On page 4, in section 3.2, it is specified that the mean BMI was 42.7. Deviation measures need to be included.
⇒ We corrected it.

Reviewer 3 Report
establishment of a national registry to track postoperative outcomes and safety is an excellent approach towards accountability and ensuring patient safety.
not entirely sure what the authors mean by "complicated input methods and a wide variety of items that 103 could not be used as models for the JSTO database". the MBSAQIP platform has data entered by a clinician other than the surgeon or their team to ensure non biased data entries. All centers enrolled use the same variables for data capture along with the same definitions to ensure homogenous data collection and collective study. leaving the each specific criteria to each facility to simplify the process does not seem a good approach towards ensuring centers use and collect the same data following surgery.
conversion rate to open seems high for laparoscopic sleeve gastrectomy. can authors specify the reason behind conversion to open procedure in these cases? also, can authors specify which organ injuries they are referring to? are these splenic injuries?
surgical site infection rate also seems elevated - are surgeons utilizing a bag to retrieve the stomach once stapled or it is being retrieved intact through the port site incision? are antibiotic prophylaxis used prior to incision?
i dont believe that out of 3240 patients only 6 - or 0.18% of patients developed GERD. The entire crux of the sleeve gastrectomy is the incidence of GERD. If this is true, authors should offer an explanation as to how their technique differs from standardized techniques, and how their GERD surveillance is done after surgery, and whether patients are treated empirically for GERD with PPI after surgery.
authors should include percentages when presenting complication rates.
the length of stay seems oddly high. some centers are already doing outpatient sleeve gastrectomies, and for the most part, sleeves should go home on postoperative day 1. can authors explain what is the cause for delay in postoperative discharge?
minor grammar issues, though the message is not distorted.
Author Response
Response to reviewer 3
We deeply thank you for your consideration and appreciate your instructive comments. We revised our article according your comments. We added descriptions and corrections.
Comments and Suggestions for Authors
establishment of a national registry to track postoperative outcomes and safety is an excellent approach towards accountability and ensuring patient safety.
not entirely sure what the authors mean by "complicated input methods and a wide variety of items that 103 could not be used as models for the JSTO database". the MBSAQIP platform has data entered by a clinician other than the surgeon or their team to ensure non biased data entries. All centers enrolled use the same variables for data capture along with the same definitions to ensure homogenous data collection and collective study. leaving the each specific criteria to each facility to simplify the process does not seem a good approach towards ensuring centers use and collect the same data following surgery.
⇒ For national registry database by JSTO, the database committee decided input items at the time of its start in . For the benefit to avoid complication at registration of each facility, it emphasizes the ease of entering input terms to simplify the process. We added description about the input methods of JSTO and the reason for simplified items(page 3,4).
conversion rate to open seems high for laparoscopic sleeve gastrectomy. can authors specify the reason behind conversion to open procedure in these cases? also, can authors specify which organ injuries they are referring to? are these splenic injuries?
⇒ The conversion rate was 36/3240 (1.1%), and no intraoperative complications nor organ damage was noted and further detail was not registered.
surgical site infection rate also seems elevated - are surgeons utilizing a bag to retrieve the stomach once stapled or it is being retrieved intact through the port site incision? are antibiotic prophylaxis used prior to incision?
⇒ A bag was used during the removal of the resected stomach in general, and prophylactic antibiotics were used immediately before incision.
i dont believe that out of 3240 patients only 6 - or 0.18% of patients developed GERD. The entire crux of the sleeve gastrectomy is the incidence of GERD. If this is true, authors should offer an explanation as to how their technique differs from standardized techniques, and how their GERD surveillance is done after surgery, and whether patients are treated empirically for GERD with PPI after surgery.
⇒ We added the description(page.4) ‘The registrations were uploaded after surgery at arbitrary timing.’ Since enrollment was voluntary immediately after surgery, some patients might develop GERD after enrollment. Postoperative prophylactic use was voluntary at each institution. It was not included in the registration items.
authors should include percentages when presenting complication rates.
⇒ We added them(page.6).
the length of stay seems oddly high. some centers are already doing outpatient sleeve gastrectomies, and for the most part, sleeves should go home on postoperative day 1. can authors explain what is the cause for delay in postoperative discharge?
⇒ In Japan, in order to confirm postoperative safety, there were many facilities where patients were discharged from the hospital in 3 to 5 days even if the surgery went smoothly yet. In the future, it is expected to be shorten. We added the description in the discussion(page.7).

Round 2
Reviewer 2 Report
In the second version of the manuscript, the comments made by the reviewers were answered. One of the facts included in the discussion was that an in-depth study of the variables collected in the database was not carried out due to the types of records carried out by the facilitators. As the authors indicate, it is difficult to perform predictive analyses, or a study of metabolic effects when standardized data collection is not followed.
For this reason, this manuscript serves as a starting point for developing data collection protocols to assess the safety and efficiency of sleeve gastrectomy performed in Japan, using the national registry by Japanese Society for Treatment of Obesity.
Focused on this idea, since an in-depth analysis of the safety and efficacy of this surgical intervention has not been carried out, it is recommended to modify the title of the work. Instead of being directed at verifying the use of these records, it would be advisable to reflect that a descriptive analysis has been carried out. Following this line, it is necessary to modify the conclusions and the abstract of the manuscript since reflecting that the registry verifies safety and efficacy is not supported by the analysis performed. As the authors indicate, it is necessary to rethink the registration methods.
Best regards
Author Response
Revise 2 : Response to reviewer 2
We deeply appreciated your consideration and instructive comments. We revised our article according to your recommendation.
Comments and Suggestions for Authors
In the second version of the manuscript, the comments made by the reviewers were answered. One of the facts included in the discussion was that an in-depth study of the variables collected in the database was not carried out due to the types of records carried out by the facilitators. As the authors indicate, it is difficult to perform predictive analyses, or a study of metabolic effects when standardized data collection is not followed.
For this reason, this manuscript serves as a starting point for developing data collection protocols to assess the safety and efficiency of sleeve gastrectomy performed in Japan, using the national registry by Japanese Society for Treatment of Obesity.
Focused on this idea, since an in-depth analysis of the safety and efficacy of this surgical intervention has not been carried out, it is recommended to modify the title of the work. Instead of being directed at verifying the use of these records, it would be advisable to reflect that a descriptive analysis has been carried out. Following this line, it is necessary to modify the conclusions and the abstract of the manuscript since reflecting that the registry verifies safety and efficacy is not supported by the analysis performed. As the authors indicate, it is necessary to rethink the registration methods.
Best regards
⇒ According to the reviewer's recommendation, we changed the title to `A descriptive analysis of sleeve gastrectomy in Japan based on national registry by Japanese Society for Treatment of Obesity’ and the conclusion to `using the JSTO database, we evaluated the indication, postoperative complications, and weight loss effect of sleeve gastrectomy in Japan.
Regarding the evaluation of the effect on preoperative comorbidities, future follow-up based on more detailed criteria was considered to be necessary. (abstract and page 7 )

Reviewer 3 Report
thanks for addressing the presented comments.
Author Response
Response to reviewer 3
We deeply thank you for your consideration.
